# A Thermal-Stable Protein Nanoparticle That Stimulates Long Lasting Humoral Immune Response

**DOI:** 10.3390/vaccines11020426

**Published:** 2023-02-13

**Authors:** Ten-Tsao Wong, Gunn-Guang Liou, Ming-Chung Kan

**Affiliations:** 1Department of Marine Biotechnology & Institute of Marine and Environmental Technology, University of Maryland Baltimore County, Baltiomre, MD 21202, USA; 2Institute of Biological Chemistry, Academia Sinica, Taipei 11529, Taiwan; 3Office of Research and Development, College of Medicine, National Taiwan University, Taipei 10051, Taiwan; 4Vaxsia Biomedical Inc., Taipei 11503, Taiwan

**Keywords:** nanoparticle, thermal stable, self-assembled protein nanoparticle

## Abstract

A thermally stable vaccine platform is considered the missing piece of vaccine technology. In this article, we reported the creation of a novel protein nanoparticle and assessed its ability to withstand extended high temperature incubation while stimulating a long-lasting humoral immune response. This protein nanoparticle was assembled from a fusion protein composed of an amphipathic helical peptide derived from the M2 protein of the H5N1 influenza virus (AH3) and a superfolder green fluorescent protein (sfGFP). Its proposed structure was modeled according to transmission electronic microscope (TEM) images of protein nanoparticles. From this proposed protein model, we created a mutant with two gain-of-function mutations that work synergistically on particle stability. A protein nanoparticle assembled from this gain-of-function mutant is able to remove a hydrophobic patch from its surface. This gain-of-function mutant also contributes to the higher thermostability of protein nanoparticles and stimulates a long lasting humoral immune response after a single immunization. This assembled nanoparticle showed increasing particle stability at higher temperatures and salt concentrations. This novel protein nanoparticle may serve as a thermally-stable platform for vaccine development.

## 1. Introduction

The thermostability of a vaccine is considered an important characteristic that is essential to fulfilling the global vaccination initiative. Current vaccine logistics practice demands a continuous cold-chain environment from the manufacturer to remote clinics to maintain vaccine efficacy. It is estimated that there are 775 million people living outside of the electrical grid [1] and beyond the reach of vaccine cold chains and vaccination, which leads to millions of children dying of vaccine-preventable diseases each year [2]. To fully cover the population beyond the electrical grid, a vaccine has to be able to withstand temperatures up to 40 °C for two months, i.e., the duration that the vaccine is stored in a health post [3].

Subunit vaccines are a safe alternative to traditional inactivated or attenuated vaccines, but their efficacy is often hindered by the low antigenicity of recombinant proteins. Different approaches have been utilized to resolve this issue; among them, virus-like particles (VLP) and self-assembled protein nanoparticles (SAPN) are considered the best platforms for subunit vaccine development [4]. The size of VLPs ranges between 20 and 200 nm, which facilitates both efficient draining to lymph node and also uptake by antigen-presenting cells such as dendritic cells and macrophages [5]. The other benefit of a VLP-based vaccine is the induction of B cell receptor clustering when repetitive antigen is presented to B cell, a function that can activate antibody class-switching and somatic hypermutation in a T-cell-dependent mechanism [6]. Not only can the virus-like particles be directly used for vaccines, but heterologous antigens can also be applied to the particle surface through genetic fusion [7]. A universal flu vaccine candidate, ectopic M2 domain (M2e), has been genetically fused with Hepatitis B core antigen (HBc) and assembled into a nanoparticle that provides full protection to heterologous flu strains [8]. VLP purified from bacterially expressed capsid protein often encloses bacterial RNA that is required for particle stability [9]. Although the enclosed bacterial RNA could potentially act as an adjuvant during immunization [10], the labile nature of RNA also contributes to the low stability of VLP. A SAPN that is assembled strictly from protein components may provide better overall vaccine stability. The SAPN based on iron-transporting protein, ferritin, has been chosen as vaccine carrier for being constituted strictly by protein subunit [11,12]. It has good thermostability when exposed to high temperatures but long-term storage data for ferritin-based vaccines have not been reported yet. Recent advancements in the computer modeling of protein structures have enabled the de novo design and assembly of protein nanoparticle from protein subunits that are known to form either pentamer or trimer [13]. When fused with an RNA-binding motif, these protein nanoparticles are able to enclose their own RNA genomes and form artificial nucleocapsid that can be used for in vitro or in vivo selection for variants with higher stability [14].

The green fluorescent protein belongs to a fluorescent protein family which is structurally conserved and emits fluorescent light from a chromophore when excited by photons of shorter wavelength [15]. The shared features of fluorescent proteins include a sturdy barrel-shaped structure constituted by 11 β-sheets and an enclosed chromophore that emits fluorescent light when excited [16]. The function of the barrel shell is to provide a well-organized chemical environment to ensure the maturation of the chromophore and protect it from hostile elements [17]. Therefore, it is conceivable that the protein sequences among fluorescent protein family members in the barrel shell are highly variable and fluorescent proteins possess desirable biophysical properties that can be selected using directed evolution [15,18,19,20,21]. The applications of fluorescent proteins have been expanded into multiple areas beyond live imaging, which includes serving as biological sensors [22,23] or detectors for protein–protein interaction or protein folding [24,25]. 

Amphipathic α-helical peptide (AHP) forms hydrophilic and hydrophobic faces when folded and is often identified in proteins related to phospholipid membrane interaction. The N-terminal amphipathic helical peptide is required for membrane anchorage of the Hepatitis C virus NS3 protein and the protease function of the NS3/NS4a complex [26,27]. Several anti-microbial peptides also possess amphipathic properties and function by forming membrane pores or causing membrane disruption [28]. The amphipathic α-helical peptide of type-A influenza virus M2 protein is required for M2 protein anchorage and induces membrane curvature required for virus budding [29,30]. This report described the identification and design of a protein nanoparticle assembled from a fusion protein composing an amphipathic α-helical peptide (AH3) from the M2 protein of type-A influenza strain H5N1 and an sfGFP. We built a protein structure model based on TEM images using Ascalaph Designer. The model suggests AH3 serves as a polymerization module, and it induces fusion protein assembly into a corn-on-a-cob structure. The sturdy barrel structure of sfGFP was explored as an antigen-presenting module for peptide antigen presentation. From the protein model, we generated a gain-of-function mutant that provides higher stability to the protein nanoparticle. We showed biophysical and immunological evidence suggesting that the protein nanoparticle assembled from a fusion protein containing a gain-of-function mutant is able to withstand extended high-temperature exposure and stimulates a long-lasting antibody response to an inserted antigen in a single immunization without adjuvant.

## 2. Materials and Methods

### 2.1. Peptide Information, Expression and Purification of Recombinant Protein

The AH3 peptide sequence is MDRLFFKCIYRRLKYGLKRG. The sequence of peptide for anti-hM2e antibody titer ELISA is SLLTEVETPIRNEWGSRSNGSSDC. The sequence of the peptide in the insertion site of sfGFP is SLLTEVETPIRNEWGSRSNGSSDSSGGSLLTEVETPIRNEWGSRSNGSSD. The protein expression vectors encoding target proteins were transformed into E coli competent cells using a heat-shock transformation. Colonies of transformed bacteria with the desired vector were scraped from the plate and inoculated in LB culture with an antibiotic. The bacterial culture was then grown exponentially to OD600 between 0.5 and 0.7 before cooling down on an ice bath and protein expression was induced by 1 mM IPTG at 20 °C for 14–16 h with 250 rpm shaking. After protein induction, bacteria were harvested by 5000 rpm centrifugation in a Sorvall SLC3000 rotor for 10 min. Bacterial pellet from 400 mL LB culture was re-suspended in 40 mL lysis buffer containing 10 mM Imidazole in 1XGF buffer (20 mM Na(PO_4_) pH7.4 and 300 mM NaCl) for sonication. For Ni–NTA resin purification, 10 mM Imidazol was added in 1XGF buffer for bacteria lysis (Lysis buffer), 20 mM Imidazol was added in 1XGF buffer for column wash (Wash buffer), and 500 mM Imidazol was added in 1XGF buffer for protein elution (Elution buffer). Bacteria were lysed using an ultrasonic sonicator (Misonix 3000) at 10 s on/20 s off cycles for 5 min at output level 5 in icy water. Insoluble cell debris was removed by centrifugation at 10,000 rpm for 10 min using a Sorval SS34 rotor at 4 °C. Soluble fraction containing the target protein was then used for purification by Ni–NTA resin as described in the user manual or was used for sedimentation ultracentrifugation. Purified proteins were stored in elution buffer in a cold room before further processing. Before immunization or thermostability testing, protein buffer was changed into 0.5× GF buffer using Sephadex-25 resin (GE, PD-10 and NAP-5). 

### 2.2. Biophysical Analysis of Protein Oligomerization and Hydrophobicity

The oligomerization state of AH3–GFP was first analyzed by protein concentration tube Vivaspin 2 with an MWCO of 100, 300, or 1000 kDa from Sartorius. Protein samples were centrifuged in Vivaspin 2 at 1000× *g* for 20 min. Filtrate was analyzed by SDS-PAGE and then coomassie blue staining was performed for analysis. TEM images of the AH3–GFP protein complex was obtained after negative staining with uranyl acetate. Images were taken using Tecnai G2 Spirit Twin. For AH3–sfGFP–2xhM2e and its variants, the purified fusion protein was first crosslinked with Sulfo-SMCC (sulfosuccinimidyl 4-[N-maleimidomethyl]cyclohexane-1-carboxylate) at 4 μg/mL for 30 min before been processed using negative staining protocol. A sucrose density gradient was used to analyze AH3–sfGFP–2xhM2e-related proteins’ oligomerization status and hydrophobicity. In a 13 mL polypropylene tube (Beckman cat#14287), the bottom was layered first with 1 mL 65% (*w*/*v*) sucrose solution and then was topped with 2 mL 45% (*w*/*v*) sucrose solution and then followed by 7 mL 15% (*w*/*v*) sucrose solution. Sudan III stock solution was prepared as 0.5% in isopropanol. Staining of the bacterial membrane was by adding Sudan III stock solution into the bacterial suspension before sonication at a 1:100 ratio. One milliliter of the soluble fraction was layered on top of 15% sucrose solution. To test protein nanoparticle membrane-binding activity, 1.5 mL baterial lysate from empty vector-transformed BL21(DE3) was mixed with 0.3 mg purified protein nanoparticle and incubated in RT for 30 min before been layered on the top of the sucrose solution and performing ultracentrifugation. The centrifuge tubes were photographed in front of a dark background under exposure to 450 nm LED light. Images were processed in ImageJ by first extracting green channels, and then the fluorescent intensity of ultracentrifugation results were quantitated from top-to-bottom using Plot Profile and compiled using a ROI manager. The protein nanoparticle’s hydrodynamic diameter was determined by dynamic light scattering (DLS) using Stunner (Unchained Lab).

### 2.3. Animal Immunization and Antibody Titer Determination

The animal protocols were approved by IACUC of Fu Jen Catholic University and mice were purchased from BioLASCO and housed in the experimental animal center of Fu Jen Catholic University following SPF standards. Mice used in immunization procedures (BALB/c strain) were between 8 and 9 weeks of age. The mice used in this study were the BALB/c strain, which is resistant to inflammation and autoimmune responses induced by adjuvant added in the vaccine. Additionally, BALB/c strain mice tend to develop Th2-type immune responses and produce higher humoral immune responses compared with C57BL/6 strain mice. Mice were immunized through an intramuscular injection of purified protein preparation in 1 mg/mL concentration. For blood collection, mice were bled from a facial vein after being pricked by the lancet. Sera collected were stored at −80 °C before being analyzed by ELISA assay. For each immunization, 20 μg of purified protein was injected intramuscularly in the thigh of the hind limb. Blood was bled 14 days post immunization or on planned dates for analysis using ELISA. For ELISA, the antigen was diluted with coating buffer for coating on a high-binding ELISA plate (Greiner Bio-One MICROLON high binding ELISA plate), and the antigen concentration of hM2e peptide was 2.5 μg/mL and sfGFP was 1 μg/mL. The ELISA plate was incubated at 4 °C overnight and then washed and blocked with 200 μL of blocking buffer containing 1% BSA in washing buffer. The washing buffer was composed of 10 mM Na(PO_4_), 150 mM NaCl at pH 7.4 with 0.05% Tween 20. Antiserum was diluted starting from 1:100 and followed by 4-fold serial dilutions. The secondary antibody, Goat anti-mouse IgG with HRP conjugation, was diluted at 1:5000 in blocking buffer. Chromogenic development was carried out by adding 100μL TMB and incubated for 10 min and stopped by adding 100 μL 2 M of H_2_SO_4_. Antibody titer was determined as the reciprocal value of the highest dilution that gives an OD450 reading of 0.1 above background.

### 2.4. Protein Structure Modeling and Intermolecular Force Calculation

Protein structure models of AH3 peptide monomer, dimer, and tetramer were generated using AscalaphDesigner version 1.8.79 (Agile Molecule, Stockholm, Sweden) and manual movement. The intermolecular interaction forces between either monomers or dimers were calculated using the intermolecular energy command of AscalaphDesigner. Hydrogen bonds between the peptide subunit and hydrophobic patch were determined and illustrated using Deep View/Swiss PDB Viewer version 4.1.1. The solvent-accessible surface area of the AH3 dimer and GFP were calculated using Jmol with a radius of 1.2 angstroms.

### 2.5. Thermostability Determination

The purified protein nanoparticle that was stored in the elution buffer was changed to 0.5× GF buffer (pH 8.0) using Sephadex-25 resin. Then the NaCl concentration of protein solutions was adjusted to the designed concentration by adding 5 M NaCl. Protein samples were incubated at various temperatures, and samples were taken at designated time points for sample preparation for SDS-PAGE and coomassie blue staining was performed at the end.

## 3. Results

### 3.1. Identification of a Protein Complex with High Antigenicity and Stability

As described in our patent application filed in 2015, we tested the immunogenicity of fusion proteins composed of an AHP and a GFP [31]. The results showed an increase of anti-GFP IgG titer in a range between 2 and 3 log under a prime and boost regime (Appendix A). One of the peptides, AH3, which is derived from the M2 protein of type-A influenza strain H5N1, provides extended stability to the GFP fusion protein when compared with another peptide, AH1 (Appendix A) as well as other peptides in our study (data not shown). Since a stable protein is essential for a vaccine carrier, we were interested in the mechanism of AH3–GFP stability and antigenicity. To study the potential mechanisms that contribute to the above-mentioned properties of AH3–GFP fusion protein, we first checked the composition of the AH3–GFP protein post expression and purification. One clue that led us to study the composition of AH3–GFP fusion protein is the difficulties encountered during protein purification. Unlike other fusion proteins studied, most of the AH3–GFP and AH5–GFP fusion proteins are expressed as insoluble inclusion body and the remaining soluble protein did not bind to Ni–NTA resin under the normal condition of 300 mM NaCl. The AH3–GFP and AH5–GFP fusion proteins only started to bind to Ni–NTA resin after we decreased the NaCl concentration from 300 mM to 50 mM. Additionally, the resistance of AH3–GFP fusion protein to hydrolysis suggested the linker between AH3 peptide and GFP is kept in a water tight complex. The likely hypothesis to explain these observations is that when GFP was fused with the AH3 peptide, the AH3 peptide induces the assembly of a hydrophobicity-driven protein complex formation that hinders N-terminal His tag from binding to Ni–NTA ligand.

### 3.2. Characterization of AH3–GFP Protein Complex

To test the hypothesis that AH3–GFP or AH5–GFP fusion protein forms a protein complex, we first used protein concentration tubes with a different molecular weight cut off (MWCO) to determine the protein complex sizes. As shown in Figure 1A, GFP protein with a molecular weight of 27 kDa was able to pass through membranes with MWCO of 100 kDa, 300 kDa, and 1000 kDa freely, but AH3–GFP fusion protein was prevented from passing through the membranes with an MWCO up to 1000 kDa. With a molecular weight of 30 kDa, the purified AH3–GFP fusion protein needs to form a complex with more than 35 monomers to be excluded from passing a membrane with a 1000 kDa MWCO. To further explore the geometric composition of the AH3–GFP protein complex, we examined the fusion protein under a transmissive electronic microscope (TEM). The TEM images showed the AH3–GFP fusion protein forms a corn-on-a-cob structure with a length up to 60 nm and a diameter around 10 nm (Figure 1B). The difference in length suggests that the particle may be assembled along the long axis. When scanning the AH3–GFP particles along their long axis, there is a repetitive pattern of the two-three–two-three array of white dots. The predicted structure, according to TEM images, is shown in Figure 1D. We also examined the AH5–GFP protein complex under TEM, but there was no clear evidence of the formation of a higher-order protein complex, suggesting the AH5–GFP protein complex is not as stable as AH3–GFP to withstand the conditions during negative staining. To find the correlation between protein complex formation and antigenicity, we immunized mice with purified AH3–GFP fusion proteins and the GFP proteins. Proteins were prepared from LPS synthesis defective *E. coli* strain, ClearColi BL21(DE3), to avoid the interference of LPS contamination, a known TLR4 ligand and an adjuvant. The mice were immunized with purified proteins by single intramuscular injection, and sera were collected at day 7, 14, 30, and 182 to evaluate anti-GFP IgG titer by ELISA. Deoxycholate was added to test if it affect in the concentration of 0.2% affects AH3–GFP antigenicity and the related experiment was terminated at 30 days post immunization when it showed no effect on the antigenicity of either GFP or AH3–GFP. These results suggested GFP alone is a poor antigen and it only gains high antigenicity after being fused with the AH3 peptide (Figure 1C).

### 3.3. Modeling the AH3–GFP Protein Complex Structure

To understand the potential molecular mechanism leading to the assembly of the AH3–GFP nanoparticle, we built an AH3 model based on a known helical structure and the hypothesis that hydrophobic interaction drives the complex formation in addition to two observations from TEM images: first, the particle was assembled along the long axis and second, the AH3–GFP protein particle had a repetitive three-two pattern along its long axis after examining the protein particle images. We first assembled two AH3 peptides as anti-parallel helices with a 24-degree angle to make close contacts involving side chains of Phe5, Phe6, Ile9, Leu13, and Leu17 using protein modeling software, AscalaphDesigner (Figure 2A). The hydrophobic core contributes the major intermolecular force binding these two helices and the AH3 dimer is surrounded by hydrophilic side chains from multiple lysine and arginine. Using another protein modeling software, Deepview, a hydrophobic patch was identified on one face of the assembled dimer as marked by a red surface (Figure 2B). In a water-accessible surface model of the AH3 dimer, two connected hydrophobic pockets can be seen located within the hydrophobic patch (marked by a dashed line) that serve as binding sites for two Arg12 side chains extruding from the opposite face of the second AH3 dimer. After turning counter clockwise for 36° while looking down the hydrophobic patch, the second AH3 dimer made close contact in tandem with the first dimer and formed a tetramer (Figure 2C). The intermolecular energy between two dimers from this model was calculated to have a ΔG of −101 kcal/mol (Figure 2C). After adding GFP protein structures onto the AH3 tetramer model, the AH3–GFP fusion protein tetramer formed a scissor-shaped assembling unit, and the stacking of every AH3–GFP tetramer on top of another tetramer increased the particle length by 2.8 nm and turned the axis by 72°. Since the GFP protein barrel diameter ranged between 2.7 and 3.5 nm, the out-extending GFP from the AH3–GFP tetramer can spatially fit into this model (Figure 2D). According to this model, the protein nanoparticle extends continuously along the long axis, with a hydrophobic patch presenting on the growing end of the assembling particle and serving as a point for polymerization.

### 3.4. Designing a Vaccine Carrier That Enables Heterologous Antigen Insertion and High Stability

After proving that the AH3–GFP protein nanoparticle possesses high antigenicity, we decided to explore the potential of the AH3–GFP protein nanoparticle as a vaccine carrier. For the Hepatitis B core antigen, the amino acid 144 may serve as an insertion site for heterologous antigen fusion and expose the antigen to the immune system [32]. The GFP protein has a thermally stable structure that is constituted by 11 β-strands and 1 α-helix, and some of the loops connecting the β-strands have been explored as insertion sites for heterologous proteins [22,23,33]. Among those candidates, loop173 linking the strand 8 and the strand 9 was chosen because it has been shown to have a high capacity for foreign peptide insertion (Figure 3A) [31]. The original AH3–GFP recombinant protein was built in pET28a vector with an AH3 coding sequence inserted into the C-terminal of His-tag and thrombin cleavage site and followed immediately by GFP cloned from pEGFP–C2. This expression vector was low in soluble protein production and unable to express soluble recombinant protein when the peptide was inserted between D173 and G174. To resolve the expression and folding issues, we designed a new expression vector. First, we cloned AH3 peptide into the N-terminal following methionine in the pET27 vector, and then to its C-terminal we inserted a 6 a.a. linker (GTTSDV) followed by a synthetic sfGFP gene [34] with an antigen insertion site next to Ser175 of sfGFP. The length of the linker between the AH3 peptide and sfGFP is important for protein nanoparticle assembly as well since the fusion protein with a 2-amino-acid linker failed to assemble into nanoparticles. The antigen insertion site also contains an 8xHis tag to facilitate protein purification. To verify vaccine carrier function, we inserted two copies of a broad-spectrum flu vaccine candidate, the M2 ectopic peptide from PR8 strain (hM2e), separated by a 6 a.a. linker (Figure 3B). The newly constructed vector was proven to be efficient for expressing soluble AH3–sfGFP–2hM2e fusion proteins as a protein complex. However, the AH3–sfGFP–2xhM2e protein complex was not as stable as AH3–GFP because these nanoparticles fell apart during distilled water washing before TEM imaging. After few trials, we found the protein nanoparticles have to be crosslinked by sulfo-SMCC before TEM analysis. Following the established AH3–GFP protein complex model, we were seeking strategies to create a more stable AH3–sfGFP protein complex. First, using the protein modeling software AscalaphDesigner, we found the mutation of Ile9 to Leu increases the intermolecular force (ΔG) between two peptide helices from −16 kcal/mol to −39 kcal/mol (Figure 3C). Second, when we mutated Lys14 to Glu14, there were two hydrogen bonds formed between LYRRLE dimers, one between the side chain of Glu14c and Cys8a and the other one the side chain of Arg12a and the oxygen of the Tyr10c backbone (Figure 3D). The intermolecular force between the LYRRLE dimers increased from −101 kcal/mol to −161 kcal/mol. The K14E mutation not only enabled tighter binding between two adjacent dimers, but also resolved a key issue of AH3-mediated protein nanoparticle assembly, an exposed hydrophobic patch. From the protein modeling results, we found that the dimer assembled from the AH3 double mutant, LYRRLE, was able to bind to pre-formed particle in two orientations: either as a tandem dimer (ΔG = −161 kcal/mol) (Figure 3F) as an inverted dimer (ΔG = −59 kcal/mol). The formation of an inverted tetramer was able to enclose the hydrophobic patch inside the protein nanoparticle (Figure 3G).

Following the protein modeling result, we went further to verify whether the protein modeling results were correct using biophysical assays. We generated mutations in the AH3 peptide sequence in the context of AH3–sfGFP–2xhM2e (Figure 4A). The exposed hydrophobic patch on the end of the AH3–GFP protein nanoparticle, according to our hypothesis, will bind bacterial membrane and the co-sediment with the membrane during the sucrose step-gradient ultracentrifugation. This methodology was first verified by loading the bacterial-soluble fraction (containing bacterial membrane) prepared from homogenized bacterial culture in a centrifuge tube preloaded with 15% (*w*/*v*), 45% (*w*/*v*) and 65% (*w*/*v*) sucrose, as demonstrated in Figure 4B’s left panel. The sedimentation of the bacterial membrane was marked by a lysochromic dye (Sudan III) that binds lipid. The control sample contained Sudan III with lysis buffer alone (Figure 4B, lane 1). After ultracentrifugation, the bacterial membrane was sedimented to the junction between 15% and 45% sucrose as marked by Sudan III staining (Figure 4B lane 2). Using the same protocol, AH3–GFP was found to co-sedimented with the bacterial membrane (Appendix A) as well as the AH3–sfGFP–2xhM2e fusion protein but not a free GFP protein (Appendix A) when the tubes were illuminated by 450 nm LED light. After protocol verification, the bacterial-soluble fractions from all six fusion protein expression cultures were prepared and analyzed using the same protocol. From the distribution of fluorescent protein between different percentages of sucrose, we made several observations: first, all of the expressed AH3–sfGFP–2xhM2e proteins bound to the bacterial membrane and were co-sedimented at the 15%/45% junction, as with AH3 variants LY and LYRLLK (Figure 4C, lane 1). Second, comparing tubes 4, 5, and 6 to tubes 1, 2, and 3, it is clear that when Lys14 was mutated to Glu, this mutation decreased the binding of expressed protein nanoparticles to bacterial membranes and about half of the protein complex remained on the top of the centrifuge tube. Third, the protein complex of AH3 variant LYRRLE and RRLE formed higher-order protein aggregates and was sedimented further to the 45%/65% junction (Figure 4C, lane 5 and 6); the identity of these high-order protein complexes is not known, but they were most likely aggregated protein nanoparticles. These results confirmed the predicted presence of a hydrophobic patch on the protein nanoparticles assembled from AH3–sfGFP–2xhM2e protein and LY, LYRLLK variants. Additionally, a portion of the protein nanoparticles assembled from AH3 variants containing a K14E mutation removed the hydrophobic patch from the surface when they were synthesized in cells. To verify if purified protein nanoparticles still have membrane-binding activity, the purified protein nanoparticle of AH3, LY, RRLE, and LYRRLE variants were first mixed with the bacterial-soluble fraction and incubated for 30 min before being evaluated by ultracentrifugation. A control tube with the same setup but no centrifugation was used to show the position of the protein nanoparticle before ultracentrifugation (Figure 4D). The data show that a fraction of the protein nanoparticles from AH3 and LY variants bound and were co-sedimented with the bacterial membrane during ultracentrifugation, as shown in the line plot that depicts the fluorescent protein distribution of all test samples (Figure 4E). None of the protein nanoparticles assembled by RRLE or LYRRLE variants co-sedimented with the bacterial membrane. These data are consistent with the protein modeling results suggesting that a protein nanoparticle assembled by the LYRRLE variant is able to eliminate hydrophobic patch through an inverted cap (Figure 3F).

To further evaluate the effect of AH3 mutants on protein nanoparticle formation, we examined the particle morphology using TEM and dynamic light scattering (DLS). The result is shown in the negative staining images (Figure 5A–D). Through visual observation and image refinement using ImageJ, we found some morphological evidence that support the protein modeling result. First, there were donut-shaped protein nanoparticles been observed in the AH3–sfGFP–2xhM2e protein preparation (Figure 5A). The donut- or disc-shape structure may be due to the collapse of the corn-on-a-cob structure because of low intermolecular interaction between AH3 monomers (ΔG= −16 kcal/mol). This hypothesis is supported by the DLS data that measure the protein particle size. These data showed a 2.5-fold increase in particle size when Ile9 (AH3) was mutated to Leu9 (LY) and an increase of predicted intermolecular force between monomers from ΔG= −16 kcal/mol to ΔG= −39 kcal/mol (Table 1). The significance of Leu9 in stabilizing particle structure is also underscored by the change in morphology of protein nanoparticles assembled by RRLD and RRLE variants. Part of the fusion proteins containing RRLD or RRLE variants assembled into an eccentric ladder-like structure, suggesting an imbalance in the interaction forces within the protein nanoparticle. 

As shown in the sucrose step gradient ultracentrifugation, the presence of a hydrophobic patch enables nonspecific interaction of the AH3–GFP protein complex with the bacterial membrane, which may restrict the free movement of protein nanoparticles and keep them from reaching draining lymph nodes for stimulating immunity [6]. To compare the antigenicity of protein complexes derived from either AH3–sfGFP–2xhM2e or LYRRLE–sfGFP–2xhM2e, we immunized mice with a single injection of either recombinant protein. According to ImageJ analysis of SDS-PAGE gel, the purified protein nanoparticles had about 80% purity. Sera were collected on days 0, 16, 50, 92, 175, and 199 following immunization to evaluate anti-hM2e and anti-sfGFP–His IgG titer by ELISA (Figure 6A,B). The geometric mean titer of anti-hM2e IgG reached the highest point for the AH3–sfGFP–2xhM2e group and then declined afterward. However, in the LYRRLE–sfGFP–2xhM2e group, the GMT reached its highest point at day 50 and remained steady up to day 90. Surprisingly, the anti-sfGFP–His antibody did not decrease in the later stages but the IgG titer increased over time until the end of the experiment (Figure 6B). When the individual mouse serum results were observed separately, only one out of five mice from the AH3–sfGFP–2xhM2e group had a higher anti-hM2e IgG titer at day 199 than at day 16. However, four out of five mice from the LYRRLE–sfGFP–2xhM2e group showed a higher antibody titer on day 199 compared with day 16 (Figure 6C). These results suggest that the two point mutations of AH3 in I9L and K14E enabled the formation of a stable, high-antigenic protein complex that stimulates long-lasting immune responses in a single immunization.

The pre-existing antibody against VLP may reduce the efficacy of booster doses due to carrier-induced epitope suppression [35]. Although GFP is a protein of low antigenicity, the fusion with AH3 strongly enhances its antigenicity, as shown in Figure 1C. To test if sfGFP backbone competes with inserted hM2e peptide for immune machinery, we immunized mice in a prime-boost protocol using the same protein preparations. The two consecutive injections were carried out 14 days apart and the sera that collected at day 14, 28, and 90 were subject to ELISA assay using either hM2e peptide or sfGFP–His protein as coating antigen. The results showed the IgG titer against hM2e elevated continuously after consecutive immunizations for both proteins as well as anti-sfGFP–His IgG titer. The result suggests that although carrier protein AH3–sfGFP also has high antigenicity, it does not interfere with the immune response against the heterologous protein, hM2e (Figure 6D,E).

One possible explanation of the long-lasting antibody response is the continuous stimulation of the immune system by the protein nanoparticles assembled by LYRRLE peptide but not AH3 peptide. To test this hypothesis, we incubated the purified protein nanoparticle in phosphate buffer containing 150 mM NaCl at 40 °C or 50 °C for 4 weeks. The integrity of protein nanoparticles was examined by SDS-PAGE and DLS. The SDS-PAGE data suggest protein nanoparticles assembled by LYRRLE and RRLE have better thermostability than those assembled by AH3 and LY peptides. The thermostability of LYRRLE-based nanoparticles is also supported by the DLS result (Figure 5G). Based on the data presented above, we conclude that the protein nanoparticle assembled by LYRRLE peptide is more suitable in serving as a vaccine carrier for two reasons. First, it is more stable in the host’s body temperature, so it can endure high-temperature storage; second, the removal of the hydrophobic patch from the surface enables the nanoparticle to move freely through vessels to reach the target site.

While working with LYRRLE protein nanoparticles, it was observed that the protein nanoparticles often degraded after the buffer was changed from high-salt elution buffer (300 mM NaCl, 500 mM Imidazole) to isotonic buffer (150 mM NaCl). This suggests a possibility of salt concentration-dependent nanoparticle stabilization. To evaluate the effect of NaCl concentration on the stability of LYRRLE-assembled protein nanoparticles, the LYRRLE–sfGFP–2xhM2e protein nanoparticles were incubated in 4 °C, 22 °C, and 37 °C for up to 3 months, and the protein stability was evaluated by SDS-PAGE. 

When the protein nanoparticles were stored at 4 °C, the protein particle stored in buffer containing 150 mM NaCl was stable during the first week, but hydrolyzed into a lower molecular weight before the fourth week (Figure 7B). Additionally, the protein nanoparticle in buffer containing 300 mM NaCl was stable for one month but it was later hydrolyzed before the end of the third month (Figure 7C). When the storage temperature was raised to 22 °C, the protein was hydrolyzed within a week in buffer containing 150 mM NaCl and within 4 weeks in buffer containing 150, 300, and 550 mM NaCl. The protein nanoparticle was only stable in buffer containing 800 mM NaCl (Figure 7D). When the temperature was shifted to 37 °C, the 300 mM NaCl was sufficient to keep the protein nanoparticle intact in the end of 3 months of incubation (Figure 7E,F). To be noted is that the protein samples that contained hydrolyzed protein nanoparticle were still fluorescent, an indication that sfGFP was still functionally intact. According to the size of hydrolysis product (20 kDa), the protein hydrolysis likely happened to the inserted loop that contains the 2xhM2e peptide. These results suggest that the stability of LYRRLE protein nanoparticle is salt dependent and it has good stability at high temperatures (37 °C). This salt-dependent nanoparticle stabilization is consistent with our assumption that the hydrophobic interaction is the major force that mediates monomer–monomer interaction.

## 4. Discussion

Vaccines as a tool to prevent infectious diseases are the most cost-effective strategy, especially for attenuated viral vaccines such as Vaccinia, MMR, or the oral polio vaccine; they produce long-lasting and even lifetime-protective immune responses, but these attenuated viral vaccines took decades to develop. Apparently, this strategy will not be able to be used to develop a vaccine in time to ward off an emerging global pandemic such as that caused by COVID-19. Although new vaccine technologies such as DNA vaccines, mRNA vaccines, or adenovirus-based vaccines can be used to quickly develop a subunit vaccine after the genomic information of a pathogen becomes available, the immune responses generated are often reduced to a baseline level within a year [36,37,38,39]. This short-lived immune response may expose vaccinated people to the risk of a breakthrough infection and the need of a booster shot. In this study, we created a self-assembled protein nanoparticle composed of a gain-of-function mutant of the AH3 peptide that enables temperature- and salt-dependent protein nanoparticle thermostability. Protein nanoparticles assembled by this gain-of-function mutant are able to stimulate a long-lasting humoral immune response that is correlated with the thermostability of the gain-of-function mutant. The surprising continued increase of anti-sfGFP IgG titer at 3 months post-immunization is distinct from anti-hM2e IgG titer. This result suggests the continued presence of functional sfGFP protein without an inserted hM2e peptide in immunized mice for 3 to 6 months. These sfGFPs were likely presented to the immune system as loop-hydrolyzed nanoparticles since individual sfGFPs have low antigenicity. The results in Figure 5G also support this possibility; when the LYRRLE–FP fusion proteins were hydrolyzed into smaller products in SDS-PAGE, the nanoparticle size was maintained. The binding and removal of water molecules from proteins in the presence of high NaCl concentrations causes an increase in hydrophobic interaction. Although the physiological NaCl concentration is maintained at 150 mM, there are other electrolytes in the body fluid that can serve the same role without disturbing the electrical potential across the cell membrane. Therefore, the in vitro stability results shown in this study may not represent what happened in injection site during immunization. The mechanism of this long-lasting immune response may be mediated by the presence of thermal stable protein nanoparticle that remains intact in the injection site and stimulates the expression of long-lasting plasma cells through continuous exposure, the same mechanism that accounts for the life-long protection of attenuated viral vaccines [40].

The fluorescent protein (FP) family is a group of proteins with a conserved, sturdy barrel structure that encloses a fluorophore. Because the function of this beta-strand-constituted barrel is to provide a tightly controlled environment for fluorophore maturation and function, directed selection against fluorescent for mutants with high tolerance to structure destabilization can improve the thermostability of the FP barrel [15]. The variability and stability of FPs make them a powerful tool as vaccine carriers. First of all, the thermostability of FP partly contributes to the thermostability of LYRRLE–FP–based protein nanoparticles, a desired vaccine property for achieving the global vaccination initiative. Second, because of the diverse origins of FP [16], the LYRRLE–FP nanoparticle platform may be expanded into a line of vaccine carriers that share a common format but each have distinct serum types. This feature of the LYRRLE–FP platform may avoid the effects of pre-existing carrier antibodies when LYRRLE–FP carrier is used in several vaccines. Third, the extensive studies on the structure of FPs and their applications have provided useful tools to incorporate heterologous antigens onto LYRRLE–FP-based protein nanoparticles. With the features mentioned above, the LYRRLE–FP format is suitable for the commercialization of a nanoparticle-based subunit vaccine.

An AH3 peptide derived from the M2 protein of type-A influenza strain H5N1 was found to induce nanoparticle formation when fused with a GFP in this study. Previous studies about the M2 amphipathic helical peptide focused on its roles in virus budding and proton pump anchorage. It mediates virus budding by generating membrane curvature by embedding its hydrophobic face in the membrane bilayer [30]. Additionally, it anchors the M2 ectopic domain on the viral envelope, which serves as a proton pump that triggers envelope membrane fusion during infection [41]. Here we discovered a novel application of AH3, which serves as a nucleating center for protein nanoparticle assembly. This novel application is restricted to conditions when an AH3 peptide is fused with a fast-folding protein such as a fluorescent protein. Fusing amphipathic helical peptide to other slow-folding proteins causes fusion protein misfolding, which appears to be due to aggregation of slow-folding fusion proteins in the presence of AH3 (unpublished results). Therefore, the fast folding ability of sfGFP is the key to AH3–FP protein nanoparticle formation. Through protein modeling, we went further to identify two point mutations in the AH3 peptide, I9L and K14E, which act synergistically to stabilize protein nanoparticles. The I9L mutation increases monomer–monomer intermolecular interaction, and the K14E mutation enables hydrogen bond formation that holds the tandem dimers together and enables the removal of hydrophobic patches from surfaces. The synergistic effect of these two AH3 mutations contributes to a more stable protein nanoparticle and restricts it from binding cellular membranes, most likely resulting in extending the lifetime of antibody responses. Although there is a limit on the size of the peptide that can be inserted into the sfGFP loop and it may restrict the scope of vaccines that this protein nanoparticle can be applied to, an alternative strategy of tethering large proteins onto the LYRRLE–sfGFP protein nanoparticle has been developed that can expand the application of this protein nanoparticle to other important pathogenic antigens (unpublished result). Even at the peptide size limit, many short peptides have been studied and found to be good vaccine candidates in addition to M2e, for example, the neoantigen for personalized tumor vaccines or peptide vaccines against various infectious diseases [42,43].

The stabilization of the LYRRLE–FP nanoparticle by a high NaCl concentration is consistent with the assumption made during structure modeling. In clinical trials involving the study of neuromuscular pain, hypertonic saline up to 5.4% (6-fold concentration of normal saline, 0.9%) is routinely used. Intramuscular injection of hypertonic saline into muscle causes intense but short-lived muscular pain with no detectable damage to the injection site tissue [44]. The use of hypertonic saline (3.6% NaCl) in aluminum hydroxide formulated antigen was able to stimulate higher cellular and humoral immune responses than normal saline in mice [45]. Other electrolytes or ingredients such as sorbitol, sucrose, histidine, or recombinant human serum albumin may be added to decrease the use of NaCl and minimize interfering membrane potential post-immunization. These electrolytes can distract water molecules from solvating proteins and keep protein nanoparticles bound by hydrophobic interaction intact. The choice of antigen can also impact the stability of LYRRLE-FP-based protein nanoparticles. An LYRRLE–FP NP that incorporated a maltose-binding protein on the surface was able to maintain both protein integrity and particle size after 2 months of incubation at 40 °C (unpublished results). The fact that hypertonic saline is safe for human administration and has the potential benefit of further boosting immune responses is encouraging to apply LYRRLE–FP protein nanoparticles for the benefits of both stimulating long-lasting immune responses and having high thermostability during storage.

## 5. Conclusions

We designed a protein nanoparticle self-assembled from a fusion protein combining two functionally distinct domains: a polymerization domain and an antigen-presentation domain. In the absence of additional adjuvant, this protein nanoparticle is able to stimulate long lasting humoral immune response in single dose. This activity is correlated with the thermostability of protein nanoparticles at 37 °C.

## Figures and Tables

**Figure 1 vaccines-11-00426-f001:**
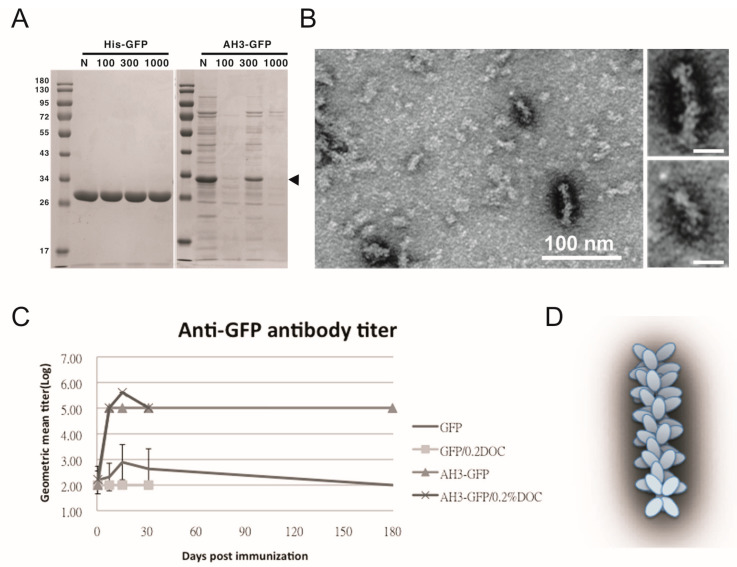
Characterization of GFP fusion protein oligomerization. (**A**) GFP and AH3–GFP fusion proteins purified and reconstituted in PBS were centrifuged through a size-exclusion membrane with a different molecular weight cut off (MWCO). The filtrate was then analyzed by 12% SDS-PAGE and coomassie blue staining. The position of AH3–GFP is marked by an arrow head. (**B**) The oligomerization of AH3–GFP protein was analyzed by negative staining using a transmissive electronic microscope (TEM). Scale bar represents 100 nm. Two of the representative particles are enlarged and shown in the right panels. (Scale bar = 20 nm) Images have been processed by Gaussian Blur filter using Photoshop. (**C**) The antigenicity of GFP and AH3–GFP fusion protein was evaluated by immunizing mice by single muscular immunization. The anti-GFP IgG titers were followed for 6 months by ELISA. (N = 5). Two samples (GFP/0.2% DOC and AH3–GFP/0.2% DOC) also included 0.2% deoxycholate when immunizing mice. (**D**) AH3–GFP protein nanoparticle model predicted from TEM images.

**Figure 2 vaccines-11-00426-f002:**
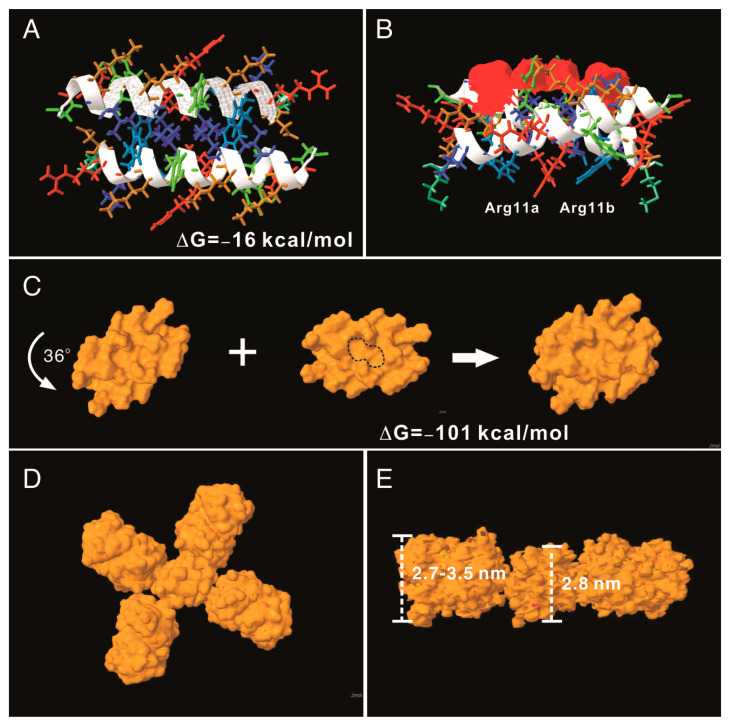
AH3–GFP protein nanoparticle structural modeling. (**A**) The assembling unit of AH3–GFP protein nanoparticles was modeled as a dimer interacting through hydrophobic interaction using AscalaphDesigner. Amino acid side chains are colored according to their hydrophobicity using SPDB v4.10. The most hydrophilic side chains are colored red, and the most hydrophobic side chains are colored blue. Side chains with hydrophobicity in between are colored transitionally. The two α-helix backbones are marked by a white ribbon. The intermolecular interaction force (ΔG) is shown in the bottom-right corner. (**B**) The bottom view of the AH3 dimer shows the location of the hydrophobic patch (marked by a red surface). The anti-parallel alpha helix fromed by two monomers forms a cross and has an angle of 24 degrees. (**C**) The water-accessible surface of the AH3 dimer is modeled using Jmol, and the stacking of the second dimer onto the first one is achieved after turning the second dimer counter clockwise by 36°. The two hydrophobic pockets are enclosed in a dashed line. The intermolecular force (ΔG) between two dimers is shown. (**D**) The AH3 tetramer with four fused GFP molecules is modeled to form a cross shape. (**E**) The front view of the AH3–GFP tetramer is shown with two GFP molecules removed for a clear view. The distance between two repeating atoms of the stacked tetramer is measured and shown as the thickness of the AH3 tetramer, and the thickness of the GFP monomer is measured as the distance across the protein barrel structure.

**Figure 3 vaccines-11-00426-f003:**
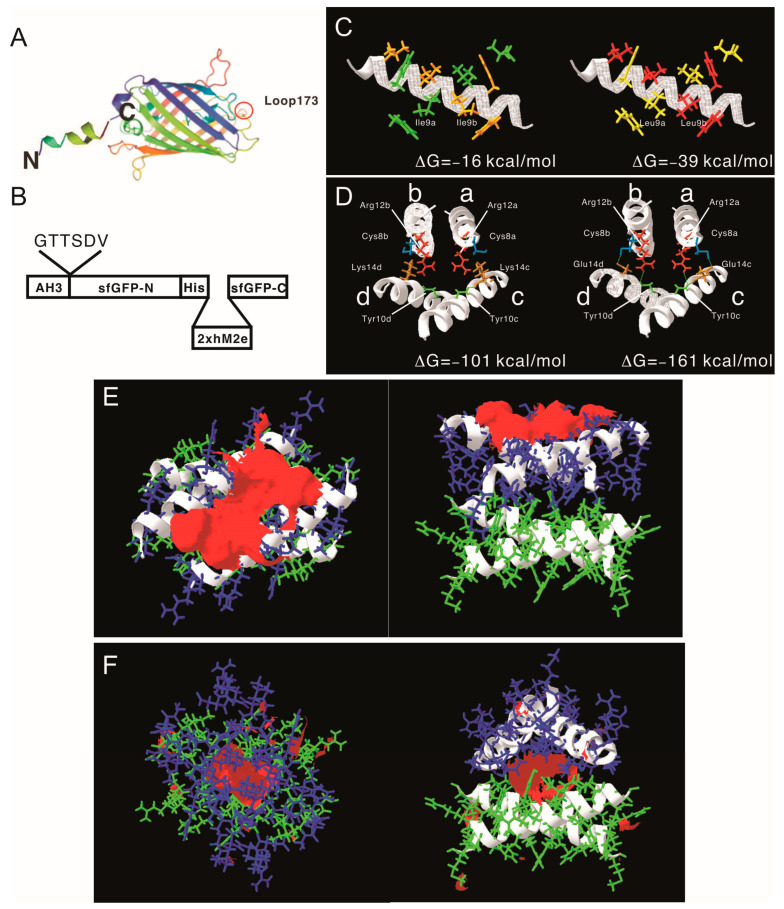
Construction of AH3–sfGFP–2xhM2e fusion protein and the protein model guided AH3 mutagenesis for variants with a higher nanoparticle stability. (**A**) The graphic presentation of the AH3–sfGFP fusion protein is shown and the antigen insertion site (loop173) is marked by a red circle. The AH3 peptide was fused to the N-terminal of sfGFP to mediate polymerization and a foreign antigen insertion site is genetically created in loop173 containing an 8xHis tag for efficient protein purification. The N-terminal and C-terminal of the fusion protein is marked by N and C respectively. (**B**) The graphic presentation of AH3–sfGFP–2xhM2e fusion protein is shown. (**C**) The comparison of AH3 and AH3 I9L (LY) mutant in dimer formation is shown. The calculated intermolecular force between two monomers are shown. (**D**) The hydrogen bonds that formed between AH3 variant LYRRLE dimers are shown in the right panel. AH3 tandem tetramer is shown in the left panel. Four monomers from each model are labeled from a to d. The Glu14 from monomers c and d form hydrogen bonds with Cys8 from monomers a and b respectively. Additionally, the side chain of Arg12 from monomers a and b forms a hydrogen bond with the backbone of monomers c and d, respectively. (**E**) The front view (left panel) and bottom view (right panel) of the LYRRLE tandem tetramer protein models. (**F**) The front view (left panel) and bottom view (right panel) of the LYRRLE inverted tetramer. The hydrophobic patch is marked as red surface, the first dimer is colored as green, and the second dimer is colored as blue. The calculated intermolecular interaction force is shown in the center of both graphs.

**Figure 4 vaccines-11-00426-f004:**
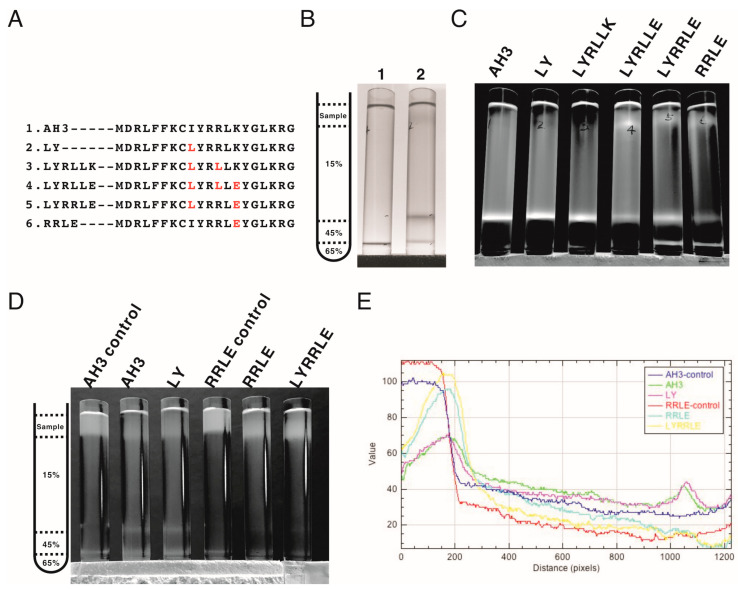
The co-sedimentation analysis of AH3–sfGFP–2xhM2e variants with bacterial membranes by analytic ultracentrifugation. (**A**) List of AH3 peptide variants introduced on the AH3–sfGFP–2xhM2e fusion protein for co-sedimentation analysis. (**B**) The right panel shows the graphic presentation of the centrifuge tube distribution of step sucrose solutions. The left panel shows the bacterial membrane distribution after ultracentrifugation, as marked by Sudan III staining. Lane 1 is a Sudan III solution only control without bacterial lysate, lane 2 is topped with 1 mL bacterial lysate mixed with Sudan III solution. The ratio between Sudan III staining solution and bacterial lysate is 1:100. (**C**) The distribution of sfGFP fusion proteins post ultracentrifugation as illuminated by 450 nm LED light was recorded as photos and analyzed by ImageJ. Only the green channel is shown. (**D**) The ability of purified protein nanoparticles to bind bacterial membranes was tested by first mixing with bacterial lysate and then analyzed by ultracentrifugation on a step sucrose gradient. The distribution of sucrose solutions of different percentages is shown in the left panel. The protein nanoparticles purified from fusion proteins containing various AH3 mutants AH3, LY, RRLE, and LYRRLE were tested. The control samples were not ultracentrifuged to reveal the original position of loaded samples. The photo taken is split into three channels by ImageJ; only the green channel is shown. (**E**) The combined line plot data showed the fluorescent intensity along the centrifuge tube from top to bottom. The image pixels had been averaged by the “smooth” command of imageJ before line plot analysis.

**Figure 5 vaccines-11-00426-f005:**
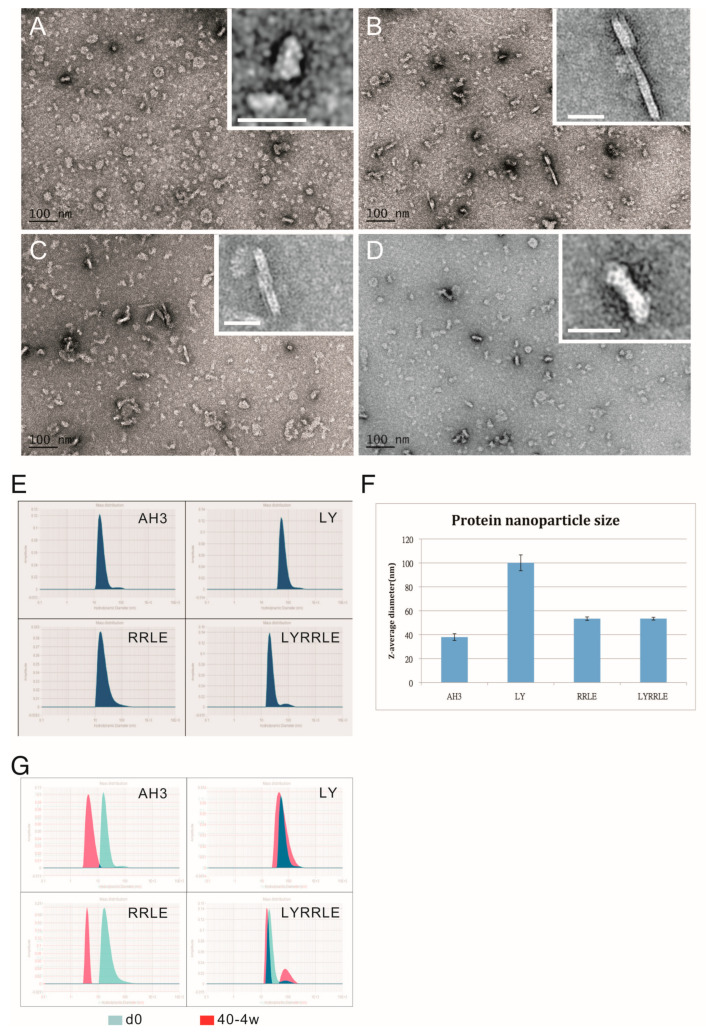
The TEM images of protein nanoparticles derived from AH3–sfGFP–2xhM2e and its variants. Purified protein nanoparticles were crosslinked with Sulfo-SMCC after buffer change from elution buffer using desalting resin (Sephadex 25). Protein samples were diluted to 0.1 mg/mL before being applied on the grid for TEM analysis. The representative images were presented in the sequence of (**A**) AH3–sfGFP–2xhM2e, (**B**) RRLD-sfGFP–2xhM2e, (**C**) RRLE–sfGFP–2xhM2e and (**D**) LYRRLE–sfGFP–2xhM2e. The enclosed window in the upper-right corner shows the magnified images of representative protein nanoparticles with size bar represent 20 nm. (**E**) The hydrodynamic diameter of protein nanoparticles derived from AH3–sfGFP–2xhM2e or its variant were analyzed by dynamic light scattering (DLS). (**F**) The Z average diameters of protein nanoparticles derived from AH3–sfGFP–2xhM2e or its variants were compared. (**G**) The merged hydrodynamic distribution graphs from protein nanoparticles prepared freshly or have been stored in 40 °C for 4 weeks.

**Figure 6 vaccines-11-00426-f006:**
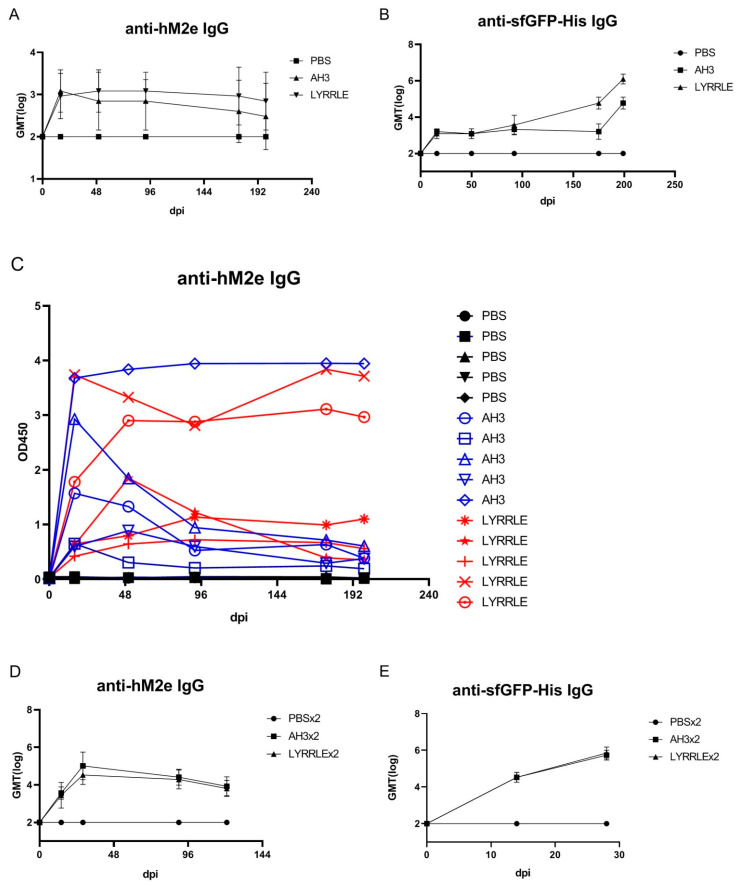
Immunization of mice with AH3–sfGFP–2xhM2e and LYRRLE–sfGFP–2xhM2e and detection of anti-hM2e IgG and anti-sfGFP IgG. The mice were immunized once at day 0 with 20 μg of purified protein nanoparticles assembled by the wild-type AH3 peptide, or the LYRRLE mutant, or PBS. Sera were collected at days 0, 16, 50, 91, 175, and 199 for analysis to detect (**A**) total anti-hM2e IgG and (**B**) total anti-sfGFP IgG by ELISA. (**C**) Anti-hM2e total IgG titers are presented as optical densities at OD450 using immune serum from each mouse diluted 1:100. (**D**) The anti-hM2e total IgG of mice immunized twice, 14 days apart, in a prime-boost regime was followed for 3 months and analyzed by ELISA (N = 5). (**E**) The anti-sfGFP–His total IgG of mice immunized twice with AH3–sfGFP–2xhM2e or LYRRLE–sfGFP–2xhM2e were evaluated 14 days after each immunization by ELISA (N = 5).

**Figure 7 vaccines-11-00426-f007:**
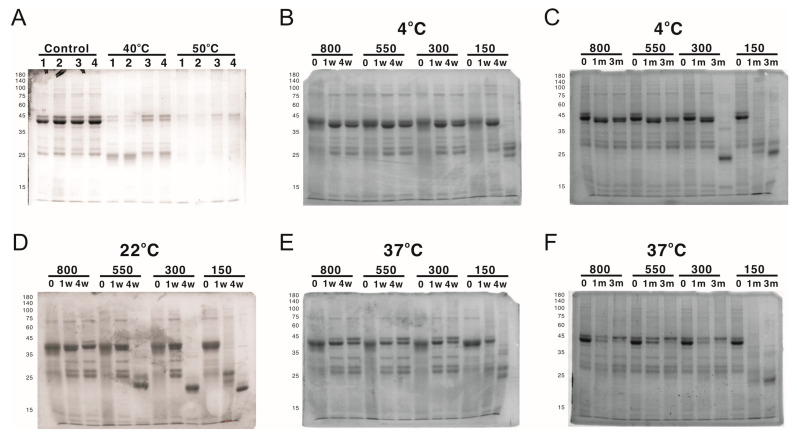
The analysis of protein nanoparticle stability using SDS-PAGE. The protein nanoparticle stability was analyzed by both SDS-PAGE and DLS. (**A**) Protein samples of 1) AH3–sfGFP–2xhM2e, 2) LY-sfGFP–2xhM2e, 3) RRLE–sfGFP–2xhM2e, or 4) LYRRLE–sfGFP–2xhM2e were incubated at either 40 °C or 50 °C for 4 weeks and analyzed in 12% SDS-PAGE for protein stability. LYRRLE–sfGFP–2xhM2e protein nanoparticles were first reconstituted in phosphate buffer containing 150 mM NaCl, and then the NaCl concentration was adjusted to either 300 mM, 550 mM, or 800 mM before being stored in 4 °C (**B**,**C**), 25 °C (**D**), or 37 °C (**E**,**F**) for the period of time indicated. Protein samples were removed and processed, and then stored before being analyzed by SDS-PAGE.

**Table 1 vaccines-11-00426-t001:** List of intermolecular forces and the access code of protein models deposited in ModelArchive.

	Assembling Unit	Intermolecular Interactions (kcal/mol)	Hydrophobic Patch Area (A^2^)	Modelarchive Access Code
AH3	Monomer	na	na	ma-06g7k
	Monomer–Monomer	−16	285	ma-bhgiw
	Tandem dimer–dimer	−101	386	ma-xhrzb
	Inverted dimer–dimer	na	na	na
LY	Monomer	na	na	ma-0koys
	Monomer–Monomer	−39	389	ma-7x5gd
	Tandem dimer–dimer	na	na	na
	Inverted dimer–dimer	na	na	na
RRLE	Monomer	na	na	ma-xkn28
	Monomer–Monomer	−23	429	ma-ax78l
	Tandem dimer–dimer	−29	342	ma-vyybf
	Inverted dimer–dimer	na	na	na
LYRRLE	Monomer	na	na	ma-ivexo
	Monomer–Monomer	−36	393	ma-lptlj
	Tandem dimer–dimer	−161	433	ma-izbbn
	Inverted dimer–dimer	−59	111	ma-xomje

## Data Availability

Protein model created in this study have been deposited in ModelArchive.org under access codes listed in Table 1.

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
