# Peer review of "A Thermal-Stable Protein Nanoparticle That Stimulates Long Lasting Humoral Immune Response"

_vaccines, 2023, doi:10.3390/vaccines11020426_

Round 1

Reviewer 1 Report

In this manuscript, Kan et al described report the development of a novel protein nanoparticle and evaluating its ability to withstand extended high temperature incubation and stimulate long lasting humoral immune response. The overall study was well designed and performed and the results are of particular interest. I suggest acceptance of this manuscript after addressing following concerns.

1.      High quality figures should be provided such as figure 1A and 1B.

2.      Scale bar for the amplification sight of figure 1B should be provided.

3.      The molecular weight for each lane should be provided for figure 7A.

4.      The immune response induced by nano-vaccines should be investigated in vitro, such as the ability to induce DC cell maturation.

5.      Potential applications of the vaccines developed in this paper need to be discussed.

Author Response

In this manuscript, Kan et al described report the development of a novel protein nanoparticle and evaluating its ability to withstand extended high temperature incubation and stimulate long lasting humoral immune response. The overall study was well designed and performed and the results are of particular interest. I suggest acceptance of this manuscript after addressing following concerns.

  1. High quality figures should be provided such as figure 1A and 1B.

Reply: We have remade the figures using coreldraw for pictures and PRISM for graphs.

  1. Scale bar for the amplification sight of figure 1B should be provided.

Reply: the scale bar is added to the figure 1B amplification sight.

  1. The molecular weight for each lane should be provided for figure 7A.

Reply: We had remade the figure 7 and added molecular weight to all pictures in figure 7.

  1. The immune response induced by nano-vaccines should be investigated in vitro, such as the ability to induce DC cell maturation.

Reply: we are planning to study the mechanisms that contributed to the long lasting humoral immunity that is produced by LYRRLE-sfGFP nanoparticle.

  1. Potential applications of the vaccines developed in this paper need to be discussed.

Reply: we have included a section in the discussion that described the potential application of protein nanoparticle discovered in this study.

Reviewer 2 Report

The manuscript submitted by Wong et al. describes the production of a protein-based vaccine delivery system. The authors describe the selection of different protein fragments for optimal stability, the manufacturing process, and preliminary observations demonstrating immunogenicity of carrier protein as well as a formulation containing influenza M2 protein fragment.

Strengths of this manuscript include rigorous evaluation of nanoparticle composition and production. Challenges that remain are limited evaluation of immunogenicity, vaccine efficacy and mechanism of action. Overall it remains difficult to evaluate how this approach compares to other formulation approaches and how it could be optimally implemented. Specific concerns are pointed out below:

Major:

- Using an LPS-negative E.coli strain is great, but most pyrogens are likely introduced during the purification process. Please include a measurement of limulus amoebocyte lysate aggregation (LAL) to definitively rule out the presence of endotoxins. This will be critical for any vaccine nanoparticle that is evaluated to eventually go into humans.

- The authors hint at purity using SDS-PAGE. Could you please include a quantitative observation pertaining to the purity of the materials?

Also, in the legend of Figure 1 (Line 229), the authors wrote “arrows are needed to point out the bands”. Presumably this is a statement that was meant to be corrected before manuscript submission, but I agree. It is currently unclear which band is what, given the large amount of apparent aggregates and breakdown products.

- In line 43 of the introduction, the authors note that one of the benefits of VLPs is the ability to induce class-switching and somatic hypermutation. However, in Figure 6, the only figure in the manuscript that alludes to immunogenicity of the VLPs produced here, only total IgG is measured. No attempts were made to investigate the isotype switching or affinity maturation of the induced antibodies.

- Line 28: “Global Vaccination Initiative” is written with capital letters,  is this referring to an organization? Please insert a reference.

- L50: bacterial RNA serves as adjuvant. This is a possibility, but given the inability of RNA and the dependence of TLR7/8 on RNA structure/length, I’d consider stating “bacterial RNA could potentially act as an adjuvant”, and add citations of studies reporting this phenomenon.

- Methods, line 142: “Mice were immunized”, there is no mention of which strain of mice is used. Not only should that be cleared up, but the authors also should include a statement pertaining to the rationale for choosing any particular strain, as the different available wild-type strains each have distinct immunological strengths and weaknesses.

-  It is unclear to the reader how the stability data presented help fulfill the global need for more stable formulations. It is my understanding that the current unmet need, as described by the authors, is for formulations that that can be stored for long periods of time, to enable access to LMIC counties. The stability data shown are not particularly impressive as they compare to currently available nanoparticle carriers. Instead, the stability data presented here appear to point more towards stability in vivo after immunization, which was not the point of this study.

- DLS was used for size measurement. I would like to see DLS used to monitor stability of the formulations over time. This will be a more commonly used, more accessible, and more comparable means of stability measurement in relation to other studies with currently licensed/used vaccine formulations.

- Line 400: “Vaccine carriers….. often failed at boosting”. Please insert references to prove this statement.

Minor:

There are many instances of incorrect grammar. Below are a few examples, but please proofread the entire manuscript:

- Line 11: please correct “evaluating” to “evaluated”

- There are three sentences in the abstract that start with the words “this protein nanoparticle”, please diversify.

- L32, “died”, consider “dying”, or “who have died”

- L35: “subunit vaccine is”, consider “subunit vaccines are”

- L37: “issue, among”. I’d start a new sentence after “issue”.

- L39; “The size…are”, correct to “The size …is’

- L45: “in the particle surface”, consider “on the particle surface”.

- L84: “we shows….evidences”, correct to “we showed….evidence”.

- L285: “may serves”, please correct to “may serve”.

- Line 444: “strong evidences”, please correct to “strong evidence”.

Author Response

- Using an LPS-negative E.coli strain is great, but most pyrogens are likely introduced during the purification process. Please include a measurement of limulus amoebocyte lysate aggregation (LAL) to definitively rule out the presence of endotoxins. This will be critical for any vaccine nanoparticle that is evaluated to eventually go into humans.

 Reply: Indeed, contamination of pyrogen usually happens during purification process when using non-pyrogenic host for protein expression. We will address this issue carefully during process development. Actually, there is another issue need to be resolved: further purification of nanoparticle after eluted from Ni-NTA resin using chromatography columns, for example the Gel-filtration column and hydrophobic column. These issues will be the focus of our future R&D.   

- The authors hint at purity using SDS-PAGE. Could you please include a quantitative observation pertaining to the purity of the materials?

Also, in the legend of Figure 1 (Line 229), the authors wrote “arrows are needed to point out the bands”. Presumably this is a statement that was meant to be corrected before manuscript submission, but I agree. It is currently unclear which band is what, given the large amount of apparent aggregates and breakdown products.

 Reply: After evaluating SDS-PAGE images using image analysis software, ImageJ, the purity of recombinant protein (LYRRLE-sfGFP-2xhM2e) used in this study has a range between 77.0% to 86.1% with an average of 82.8%. We have marked the position of AH3-GFP in Figure 1A using an arrowhead.   

- In line 43 of the introduction, the authors note that one of the benefits of VLPs is the ability to induce class-switching and somatic hypermutation. However, in Figure 6, the only figure in the manuscript that alludes to immunogenicity of the VLPs produced here, only total IgG is measured. No attempts were made to investigate the isotype switching or affinity maturation of the induced antibodies.

 Reply: We do have evidences of affinity maturation and isotype switching according to phage display selection and sequencing of antibodies induced by LYRRLE-sfGFP based protein nanoparticles with a different antigen. But these data are not the focus of this study. This statement is to express the significance of protein nanoparticle in general, not specifically to this study.

- Line 28: “Global Vaccination Initiative” is written with capital letters,  is this referring to an organization? Please insert a reference.

 Reply: Yes, we meant to describe the general efforts from several organizations to expand the vaccination coverage in LMIC. So these words should not use capital letters. We have changed them.

- L50: bacterial RNA serves as adjuvant. This is a possibility, but given the inability of RNA and the dependence of TLR7/8 on RNA structure/length, I’d consider stating “bacterial RNA could potentially act as an adjuvant”, and add citations of studies reporting this phenomenon.

 Reply: Thanks for reminding us about the missing citation regarding to the adjuvant effect of bacterial RNA enclosed in VLP. We also agree on the structural factor in the activation of TLR7/8 and make adjustment on the wording related to bacterial RNA on innate immunity. We have added the reference: C Gomes A, Roesti ES, El-Turabi A, Bachmann MF. Type of RNA Packed in VLPs Impacts IgG Class Switching-Implications for an Influenza Vaccine Design. Vaccines (Basel). 2019 Jun 4;7(2):47. doi: 10.3390/vaccines7020047. PMID: 31167472; PMCID: PMC6630894.

- Methods, line 142: “Mice were immunized”, there is no mention of which strain of mice is used. Not only should that be cleared up, but the authors also should include a statement pertaining to the rationale for choosing any particular strain, as the different available wild-type strains each have distinct immunological strengths and weaknesses.

 Reply: The mice been used in this study are BALB/c strain, which is resistant to inflammation and autoimmune responses induced by adjuvant added in the vaccine. Also BALB/c strain mice tend to develop Th2 type immune response and produce higher humoral immune response compared to C57BL/6 strain mice. This statement is added into methods.

-  It is unclear to the reader how the stability data presented help fulfill the global need for more stable formulations. It is my understanding that the current unmet need, as described by the authors, is for formulations that that can be stored for long periods of time, to enable access to LMIC counties. The stability data shown are not particularly impressive as they compare to currently available nanoparticle carriers. Instead, the stability data presented here appear to point more towards stability in vivo after immunization, which was not the point of this study.

 Reply: There are several factors control the stability of protein nanoparticle in solution. One of them is the zeta potential of particles. The isoelectric point of sfGFP is pH 6.0 and the antigen peptide including His-tag that are added on the sfGFP loop is about pH 6.0 as well, which makes the particle tends to aggregate in physiological buffer (pH 7.4). We have incorporated a antigen with theoretical pI of pH 5.08 onto the LYRRLE-sfGFP based nanoparticle and found 85% protein remained intact after 40oC for 2 months (unpublished result). Other factor like antigen structure may also impact on analysis of nanoparticle stability using SDS-PAGE since hM2e is not structured and the peptide bond is prone to hydrolysis and produces smaller fragments. In this study, we intend to establish the correlation between salt concentration and LYRRLE based protein nanoparticle stability and show an easy and low cost strategy to stabilize protein nanoparticle. And this study shows a protein nanoparticle with higher in vitro stability may contribute to durable humoral immunity. The mechanisms will be explored in our future study. 

- DLS was used for size measurement. I would like to see DLS used to monitor stability of the formulations over time. This will be a more commonly used, more accessible, and more comparable means of stability measurement in relation to other studies with currently licensed/used vaccine formulations.

 Reply: We have included the one-by-one comparison of the DLS result before and after 4 weeks incubation and presented the data as Figure 5G and has included descriptions in Result and figure legend. 

- Line 400: “Vaccine carriers….. often failed at boosting”. Please insert references to prove this statement.

 Reply: We have revised the wording to fit with scientific findings about carrier antibody interference against VLP immunization and added a reference support this statement.

Minor:

There are many instances of incorrect grammar. Below are a few examples, but please proofread the entire manuscript:

- Line 11: please correct “evaluating” to “evaluated”

- There are three sentences in the abstract that start with the words “this protein nanoparticle”, please diversify.

- L32, “died”, consider “dying”, or “who have died”

- L35: “subunit vaccine is”, consider “subunit vaccines are”

- L37: “issue, among”. I’d start a new sentence after “issue”.

- L39; “The size…are”, correct to “The size …is’

- L45: “in the particle surface”, consider “on the particle surface”.

- L84: “we shows….evidences”, correct to “we showed….evidence”.

- L285: “may serves”, please correct to “may serve”.

- Line 444: “strong evidences”, please correct to “strong evidence”.

Reply: we have made corrections to the above mentioned typos and grammar mistakes. Also we have proof reading the manuscript using Quillbot, an AI assisted grammar check service.

Reviewer 3 Report

This article has developed a new type of heat-resistant and salt-tolerant nanoparticle platform through fusion protein composed of an epidemic 13 helio-peptide derived from M2 protein of H5N1 influenza virus (AH3) and a super folder green fluorescence protein (sfGFP) to solve the heat-resistant problem of the epidemic vaccine, which has certain novelty.

Problem 1: The B and C icons in Figuer 1 are not recorded correctly. Please correct them.

Problem 2: The structure identified by TEM in figure 1 is not very similar to the structure model of figure 2. Can we further identify the AH3-GFP protein nanoparticle structure characterized in figure 1 by using other methods such as freeze electron microscopy.

Problem 3: The unstable Hm2e structure can be transformed into stable nanoparticle structure by inserting short peptides and reducing free energy through molecular simulation software in figure 3. These structures have been characterized in figure 5, but the size difference of nanoparticles is large and the morphology is also uneven. Can we further characterize this structure by using immunoelectron microscopy or cryoelectron microscopy

Problem 4: The immune test results in figuer 6 show that the picture quality is very poor, so you can use software such as PRISM to redraw the picture. The antibody titers of the other two structures are not significantly different. Whether the neutralizing antibody and other data can be added to support the functional verification of their humoral immune response.

Problem 5: figuer 7, simple vaccine shelf life test should increase the effect evaluation after animal immunization.

Problem 6: The change of protein concentration should be increased by gray analysis and BCA detection. To quantify the change of protein concentration due to temperature change, so as to more intuitively observe the thermal stability of the vaccine.

General problems: 

1. It is necessary to thoroughly update the quality of the article pictures to ensure that the publication level is reached. 2. Add characterization data to ensure correct structure. 3. The research progress of nanoparticles in thermal stability should be added in the introduction.

Author Response

Problem 1: The B and C icons in Figuer 1 are not recorded correctly. Please correct them.

Reply: We have corrected the issue. 

Problem 2: The structure identified by TEM in figure 1 is not very similar to the structure model of figure 2. Can we further identify the AH3-GFP protein nanoparticle structure characterized in figure 1 by using other methods such as freeze electron microscopy.

Reply: The figure 2 model only shows the polymerization module (AH3 peptide, 20 a.a.) when assembled as dimer and or tetramer, not the whole particle. The whole particle may contain up to 10 tetramers. In figure 2, the GFP structure is shown in figure 2D (as tetramer) and 2E (two GFP structures removed from a tetramer). More examples of nanoparticle TEM images can be found at Fig.5. 

Problem 3: The unstable Hm2e structure can be transformed into stable nanoparticle structure by inserting short peptides and reducing free energy through molecular simulation software in figure 3. These structures have been characterized in figure 5, but the size difference of nanoparticles is large and the morphology is also uneven. Can we further characterize this structure by using immunoelectron microscopy or cryoelectron microscopy

Reply: Since the major force mediating nanoparticle formation is the hydrophobic interaction, the distill water wash step during TEM sample preparation disrupts particle integrity. Although we have solved this problem using chemical crosslinker, but the crosslinking efficiency apparently did not reach 100% and many disrupted particles shows up during sample preparation. Nonetheless, the DLS data provided in figure 5E suggest these particles distributed in a small range, which is expected for protein nanoparticle assembled in one axis. We agree that cryo-EM will be the best approach to capture the full structural detail of this particle.

Problem 4: The immune test results in figuer 6 show that the picture quality is very poor, so you can use software such as PRISM to redraw the picture. The antibody titers of the other two structures are not significantly different. Whether the neutralizing antibody and other data can be added to support the functional verification of their humoral immune response.

Reply: We have updated the graphs in figure 6 with PRISM for better quality. In the figure 6C, we meant to demonstrate the durability of antibody titer when the protein nanoparticle is assembled by LYRRLE variant. Unfortunately, we do not have the capacity to carry out assay to determine neutralizing antibody titer. 

Problem 5: figuer 7, simple vaccine shelf life test should increase the effect evaluation after animal immunization.

Reply: We believe the effect of high salt in stabilizing LYRRLE based protein nanoparticle is representing in vivo conditions after immunization, where body fluid contains multiple electrolytes that are able to deprive water molecules and keep them from dissociating fusion protein from nanoparticle.  

Problem 6: The change of protein concentration should be increased by gray analysis and BCA detection. To quantify the change of protein concentration due to temperature change, so as to more intuitively observe the thermal stability of the vaccine.

Reply: We have evaluated the nanoparticle integrity using DLS after 40oC incubation and compared them with the control group (before heat treatment). The data is shown as Figure 5G.

General problems: 

  1. It is necessary to thoroughly update the quality of the article pictures to ensure that the publication level is reached.

Reply: Thanks for the comment, we have made improvements regarding to pictures quality in figure 1, 2, 3, 5, 6, and 7.

  1. Add characterization data to ensure correct structure.

Reply: We will use cryo-EM to further characterize nanoparticle structure in our future study.

  1. The research progress of nanoparticles in thermal stability should be added in the introduction.

Reply: we have included a section in the introduction that described the progress in thermostable protein nanoparticle development.

Round 2

Reviewer 1 Report

This article has been revised to meet the publication criteria, but please note some issues

1. Please confirm the spelling in this article carefully.

2. Please note the resolution of the figures in this paper

Reviewer 2 Report

Thank you for your adequate response to my concerns.